# Flustramine Q, a Novel Marine Origin Acetylcholinesterase Inhibitor from *Flustra foliacea*

Natalia M. Kowal [1], Xiaxia Di [1,2], Sesselja Omarsdottir [1] and Elin S. Olafsdottir [1,*]

[1] Department of Medicinal and Natural Products Chemistry, Faculty of Pharmaceutical Sciences, School of Health Sciences, University of Iceland, 107 Reykjavik, Iceland

[2] Department of Biochemistry & Biomedical Sciences, Michael G. DeGroote Institute for Infectious Disease Research, McMaster University, Hamilton, ON L8S 4L8, Canada

\* Correspondence: elinsol@hi.is; Tel.: +354-525-5804

**Abstract:** The bryozoan *Flustra foliacea* produces a range of indole alkaloids, and some have shown weak antibiotic, muscle-relaxant and cytotoxic properties; however, most of them have not been tested for bioactivity. Many of these alkaloids possess a physostigmine scaffold, and physostigmine is a well-known acetylcholinesterase (AChE) inhibitor. AChE inhibitors are of interest as drug leads in neurodegenerative diseases and are currently used in symptomatic treatment of Alzheimer's disease (AD). In this study, the AChE inhibitory activity of Flustra alkaloids was studied in vitro using the colorimetric method of Ellman and AChE from *Electrophorus electricus*. Twenty-five compounds isolated from the Icelandic bryozoan *F. foliacea* were screened at a 100 μM concentration. Two of them, flustramine E and flustramine I, showed inhibition of 48%, and flustramine Q showed 82% inhibition. For flustramine Q, the $IC_{50}$ was 9.6 μM. Molecular modelling and docking studies indicated that simple in silico designed derivatives of flustramine Q could have potential for increased potency. Marine natural products including brominated indole alkaloids from *Flustra foliacea* are an interesting new source of AChE inhibitors with potential towards central nervous system disorders, e.g., Alzheimer's disease.

**Keywords:** acetylcholinesterase inhibitor; flustramine Q; *Flustra foliacea*; marine natural products; *Flustra foliacea* alkaloids; marine acetylcholinesterase inhibitors; Alzheimer's disease

## 1. Introduction

The enzyme acetylcholinesterase (AChE) is a drug target of interest for neurodegenerative diseases such as Alzheimer's disease (AD) [1,2]. AChE is responsible for rapid degradation of the neurotransmitter acetylcholine (ACh), which activates nicotinic acetylcholine receptors and facilitates synaptic neurotransmission [3]. According to the cholinergic hypothesis developed in the 1980s [4–6], increasing and preserving the amount of ACh at a synapse can improve neurotransmission, and compounds that inhibit AChE have been used extensively for the last decades as drugs to manage symptoms of AD [2,7]. However, there is a need for new drug leads and structures with improved activity profiles [7,8]. A wide range of compounds of plant origin, including a big group of alkaloids, show AChE inhibitory activity [9–12]. Figure 1 shows the structures of six well-known AChE inhibitors that have been used clinically for AD or as lead compounds. Tacrine was first on the market in the USA in 1993 but is now withdrawn due to adverse effects [13]. Donepezil, rivastigmine and galantamine are currently registered drugs and used for symptomatic treatment of AD [8,14]. Galantamine is a natural alkaloid isolated from the common snowdrop (*Galanthus nivalis*), approved for mild-to-moderate AD in 2004 [15]. Donepezil is a synthetic piperidine-based molecule widely used for AD since 1996 [16]. Physostigmine, an indole-type alkaloid from the Calabar bean, served as a lead structure for the semisynthetic AD drug rivastigmine [17]. Huperzine A is a lycopodium alkaloid with strong AChE-inhibiting activity found in *Huperzia* plant species [10,18].

**Figure 1.** Structures of well-known AChE inhibitors.

Since the exploration of marine natural products (MNPs) began in the 1970s, they have proven to be a rich resource and inspiration for drug design, and several compounds were introduced to clinics, including ziconotide (Prialt, 2004) used in severe chronic pain and several anticancer agents such as cytarabine (1969), trabectedin (Yondelis, 2007) and eribulin (Halaven, 2010), a simplified analogue of the marine natural product halichondrin B. Numerous types of living sea organisms have been explored including bacteria, fungi, algae, sponges, corals, bryozoans, molluscs, tunicates, echinoderms and mangroves [19,20] and have shown a spectrum of different activities. Marine natural products often possess anticancer, anti-inflammatory, analgesic, antiviral and antibacterial properties [21]. Neu-rodegenerative diseases were not of particular interest of the marine-sourced molecules; however, a range of compounds have been discovered with cholinesterase activities [22], including several alkaloids with potential anti-AChE activities [22,23], such as barettin [24], petrosamine [25] and fascaplysin [26]. Echinochrome A, clinically used to treat acute coro-nary syndrome, has been proved as a powerful AChE inhibitor [23,27]. However, the usefulness of AChE inhibitors of marine origin as potential drug leads needs to be explored. Alkaloids recently isolated from *Flustra foliacea* by Di et al. [28] contain physostigmine-like skeletons; therefore, they are interesting to explore as potential AChE inhibitors. In this study, twenty-five alkaloids were tested for AChE inhibitory activity (Figure 2), includ-ing 23 brominated alkaloids, i.e., flustramines Q, R, T, V and W (**1–5**), flustraminols C–H (**6–11**), *N*α-methyldeformylflustrabromine (**12**), 6-bromo-*N,N*-dimethyltryptamine (**13**), 2-[6-bromo-1*H*-indol-3-yl]-*N*-methylethanamine (**14**), deformylflustrabromine (**15**), flus-tramine L (**16**), dihydroflustramine C (**17**), flustramines A, E, B, I, C (**18–22**), 6-bromoindole-3-carbaldehyde (**23**) and two imidazole alkaloids, flustrimidazoles A and B (**24** and **25**) [28]. We identified one new AChE inhibitor not described previously, flustramine Q, which presents a new scaffold in the AChE inhibitors family. Molecular modelling studies re-vealed the binding mode of flustramine Q, and two in silico analogues with predicted increased activity were designed and docked into the AChE crystal structure.

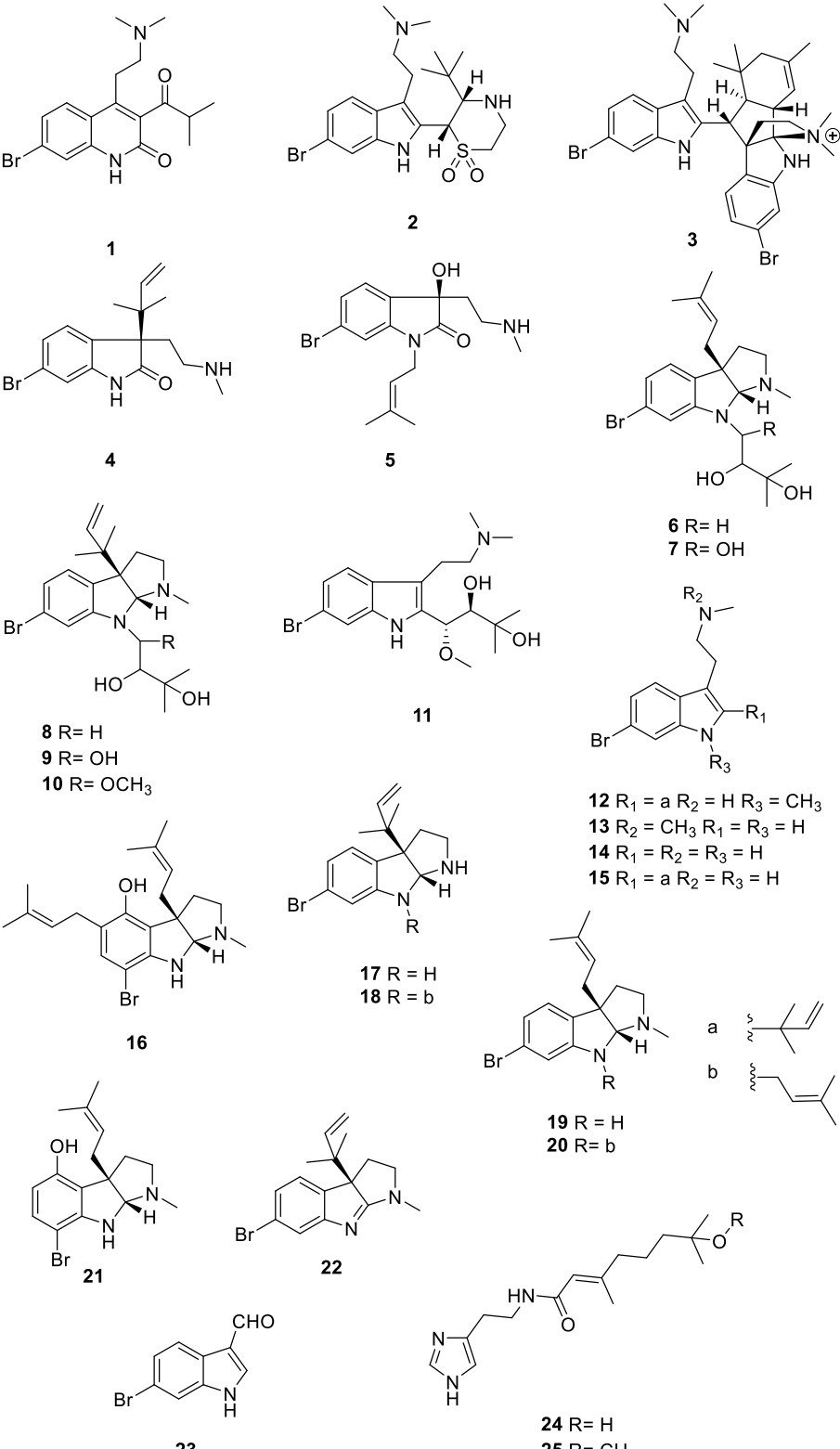

**Figure 2.** Structures of Flustra alkaloids tested against AChE activity. Flustramine Q (**1**), flustramine R (**2**), flustramine T (**3**), flustramine V (**4**), flustramine W (**5**), flustraminol C (**6**), flustraminol D (**7**), flustraminol E (**8**), flustraminol F (**9**), flustraminol G (**10**), flustraminol H (**11**), *Nα*-methyldeformylflustrabromine (**12**), 6-bromo-*N,N*-dimethyltryptamine (**13**), 2-[6-bromo-1*H*-indol-3-yl]-*N*-methylethanamine (**14**), deformylflustrabromine (**15**), flustramine L (**16**), dihydroflustramine C (**17**), flustramine A (**18**), flustramine E (**19**), flustramine B (**20**), flustramine I (**21**), flustramine C (**22**), 6-bromoindole-3-carbaldehyde (**23**), flustrimidazole A (**24**), flustrimidazole B (**25**).

## 2. Materials and Methods

### 2.1. Validation of Activity at AChE

Twenty-five alkaloids previously isolated from *F. foliacea* by Di et al. [28] were tested in vitro against AChE inhibitory activity. Their structures are numbered 1–25 and are shown in Figure 2.

The colorimetric method of Ellman [29] was used to study the in vitro AChE inhibition by the alkaloids. The AChE enzyme used for the assay was obtained from *Electrophorus electricus* (purchased from Sigma-Aldrich Co. LLC). The test solutions were applied to a 96-well microplate along with 0.2 mg/mL bovine serum albumin, 0.5 mM 5,5-dithiobis-(2-nitro-benzoic acid) and 0.05 mM acetylthiocholine iodide (ATCI). To initiate the enzymatic reaction, 0.20 U/mL of the AChE enzyme was added to each well followed by the colorimetric detection at 405 nm. Experiments were conducted in triplicate. All compounds were dissolved in methanol (max. 2% methanol at assay conditions which did not affect the enzyme activity), the test solutions were prepared at 100 μM concentration and screened using physostigmine (Eserine E8375, Sigma-Aldrich) as a positive control. AChE enzyme and all other chemicals required for the assay were purchased from Sigma-Aldrich Co. LLC (St. Louis, MO, USA) and were of analytical grade. For two compounds exhibiting the highest inhibition of ATCI degradation at 100 μM, $IC_{50}$ values were determined using 6 concentrations within the 20–90% inhibition range.

### 2.2. Data Analysis

The AChE inhibition of each test solution in the assay was evaluated by plotting the absorption readings versus time, followed by linear regression. Inhibition percentages were calculated from the obtained slopes, and $IC_{50}$ values were determined from non-linear regression calculations with GraphPad Prism version 9.4.0 (GraphPad Software, Inc, La Jolla, CA, USA).

### 2.3. Molecular Modelling

Docking of flustramine Q and its derivatives was performed with Schrödinger software package version 2021-3 (Schrödinger LCC, Portland, OR, USA). A crystal structure of a recombinant human acetylcholinesterase complexed with huperzine A (PDB ID: 4EY5 [30]) was downloaded from the Protein Data Bank (https://www.rcsb.org) and prepared following a standard Protein Preparation Workflow [31] in the Schrödinger's software. Flustramine Q and its derivative structures were converted into 3D models, and protonation states at pH = 7.4 were predicted using LigPrep module (Schrödinger Release 2021-3: LigPrep, Schrödinger, LLC, New York, NY, 2021). A low-energy conformation was found for each compound using Macro Model [32], and then ligand docking was performed using Glide [33]. Obtained docking results were visually inspected and interactions were measured in PyMOL.

## 3. Results

### 3.1. In Vitro Activity of Alkaloids from Flustra foliace

In the work presented here, 25 alkaloids isolated from *F. foliacea* were tested for inhibitory activity against AChE from electric eel. Initially, all compounds were screened at a concentration 100 μM, and two of them, flustramine Q (**1**) and flustramine I (**21**), showed inhibitory activity of 82% and 48%, respectively. The $IC_{50}$ values for these two compounds were determined to be 9.6 and 113 μM, respectively (Table 1).

**Table 1.** In vitro activities of selected compounds from *Flustra folicea*. All compounds were evaluated at 100 μM. Compound 1 and 21 as well as the control compound physostigmine were evaluated for $IC_{50}$ values.

| Compound Name | AChE %Inhibition $IC_{50}$ (μM) $pIC_{50} \pm$ SEM | Compound Name | AChE %Inhibition $IC_{50}$ (μM) $pIC_{50} \pm$ SEM |
|---|---|---|---|
| **Physostigmine \*** | $IC_{50} = 0.78$ $(6.13 \pm 0.02)$ | **12** | $17.8 \pm 4\%$ |
| **1** Flustramine Q | $\mathbf{82.1 \pm 1\%}$ $\mathbf{IC_{50} = 9.6 \pm 4}$ | **13** | $43.3 \pm 2\%$ |
| **21** Flustramine I | $\mathbf{47.9 \pm 6\%}$ $\mathbf{IC_{50} = 113 \pm 8}$ | **14** | $32.7 \pm 3\%$ |
| **2** Flustramine R | $28.4 \pm 1\%$ | **15** Deformylflustrabromine | $44.2 \pm 4\%$ |
| **3** Flustramine T | $31.5 \pm 4\%$ | **16** Flustramine L | $9.7 \pm 3.4\%$ |
| **4** Flustramine V | $26.9 \pm 2\%$ | **17** Dihydroflustramine C | $31.7 \pm 3\%$ |
| **5** Flustramine W | $24.4 \pm 1\%$ | **18** Flustramine A | $21.7 \pm 6.8\%$ |
| **6** Flustraminol C | $23.2 \pm 2\%$ | **19** Flustramine E | $47.8 \pm 2\%$ |
| **7** Flustraminol D | $17.2 \pm 2\%$ | **20** Flustramine B | $43.0 \pm 3.2\%$ |
| **8** Flustraminol E | $31.3 \pm 12\%$ | **22** Flustramine C | $19.2 \pm 4.7\%$ |
| **9** Flustraminol F | $9.1 \pm 2\%$ | **23** | $11.7 \pm 1.5\%$ |
| **10** Flustraminol G | $17.7 \pm 4\%$ | **24** Flustrimidazole A | $25.2 \pm 3\%$ |
| **11** Flustraminol H | $23.5 \pm 2\%$ | **25** Flustrimidazole B | $3.8 \pm 1\%$ |

\* Literature value of $IC_{50}$ for physostigmine from human erythrocyte AChE is 27.9 nM [34], which corresponds well to the value measured in our assay.

### 3.2. Docking of Flustramine Q into AChE Crystal Structure

Flustramine Q (**1**), which does not have an indole ring structure and is different to the physostigmine scaffold, showed high, single-digit micro-molar inhibitory activity towards AChE. This was an interesting, yet unexpected outcome. To understand the activity of flustramine Q at AChE and to analyse its possible interactions with the target, flustramine Q was docked into a crystal structure of human recombinant AChE. Flustramine Q was docked into AChE with a considerably high docking score of −9.384, as could be expected knowing it has in vitro activity. Docking results revealed that flustramine Q creates multiple hydrophobic interactions with the active site of AChE, including the aromatic residues, Tyr337 and Trp86, known to be important for the binding of known inhibitors [3,35]. It also creates a cation-π interaction with Tyr337 through a quaternary nitrogen cation and furthers its aromatic rings as well as the aliphatic part of the structure, creating hydrophobic contacts with Trp86. Additionally, flustramine Q forms a hydrogen bond with Tyr133, a weak hydrogen bond with the carbonyl oxygen of Trp86 and a halogen bond with Asp74 (Figure 3A).

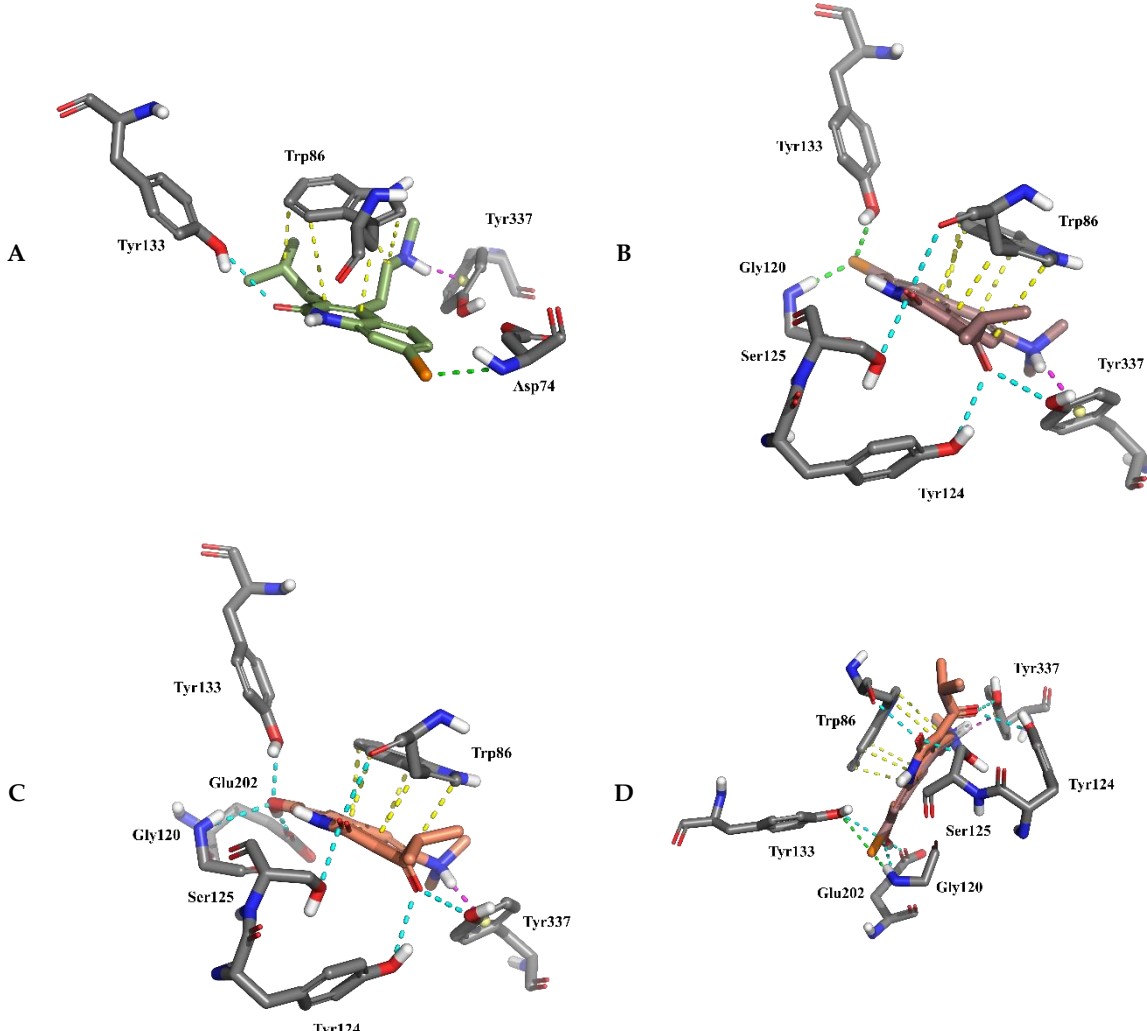

**Figure 3.** Docking results of flustramine Q and its derivatives. (**A**) Flustramine Q (moss green) docked into a crystal structure of AChE (PDB ID: 4EY5), (**B**) Flu Q Der I (mauve purple) docked into AChE, (**C**) Flu Q Der II (salomon pink) docked into AChE, (**D**) overlay of docked poses of Flu Q Der I and Flu Q Der II. Interactions of the compounds with the binding pocket of AchE are shown, and for the clarity of the figures, only interacting amino acids are displayed (slate gray). Interactions are shown as dotted lines; hydrpohobic interactions are in canary yellow, cation-π in magenta, hydogen bonds in cyan and halogen hydrogen bonds in grass green. The bromine atom is marked in tangerine orange.

### 3.3. Docking of Flustramine Q In Silico Designed Derivatives into AChE Crystal Structure

The *N*-cation-containing part of the molecule is located at the end of an aliphatic flexible chain which, in addition to its flexibility, is prone to form internal hydrogen bonds. These two features suggest that optimal interactions might not be maintained; therefore, a modification of the flustramine Q structure aiming for rigidification of the tertiary nitrogen might be beneficial for the activity. The simplest way to fix atoms in a desired position is by locking them in a ring. In the case of flustramine Q, connecting carbons 4 and 14 leads to the creation of an additional cyclohexane ring (Figure 4); however, this also introduces stereochemistry to the structure (and did not improve docking score, data not shown). A better approach is to further add a double bond which will make the whole structure more rigid and deprived of conformational freedom (Figure 4). This derivative of flustramine Q (Flu Q Der I) was designed in silico and docked into the same AChE crystal structure. The docking revealed a better docking score (from −9.38 to −10.70) and improved interactions with the AChE binding pocket (Figure 3B). Flu Q Der I shows improved interactions with

the AChE binding pocket by creating more contacts with the active site. The improved interaction profile includes more hydrophobic contacts as well as additional, stronger hydrogen bonds and a halogen bond with the two amino acids Tyr133 and Gly120. A second, potentially more active, derivative of flustramine Q (Flu Q Der II) was designed in silico by replacing the bromine atom with a hydroxyl group and docked into AChE (Figure 3C). As expected, the docking score was improved from −9.38 to −11.48 compared to flustramine Q and −10.70 to −11.48 compared to Flu Q Der I. The docked pose remained the same as Flu Q Der I (Figure 3D), but the –OH group which replaced the bromine atom creates stronger interactions with the active site through three hydrogen bonds to Glu120, Tyr133 and Glu202. Our docking studies suggest that simple derivatives of flustramine Q with increased inhibitory activity towards AChE can be designed and could be considered as future drug leads.

**Figure 4.** Structures of flustramine Q and its in silico derivatives.

## 4. Discussion

Flustramine Q (**1**) is a novel compound isolated recently from Icelandic bryozoan *F. foliacea* [28], and now its high AChE inhibitory activity is reported for the first time with an $IC_{50}$ value of 9.6 µM. Interestingly, the structure of flustramine Q is different from the physostigmine skeleton. It contains quinolin-2-one ring instead of an indole moiety, and the third ring present in physostigmine is missing (Figures 1 and 2). The unprecedented rearranged prenylated oxindole alkaloid flustramine Q [28] contains the quinolin-2-one ring, which could be derived from the known brominated alkaloid flustramine E, undergoing multiple transformations, including methylation, 1,2-alkyl shift, oxidation and dehydration to forge the quinolinone ring system [36,37]. This natural quinoline of marine origin preserves some structural features of the known synthetic AChE inhibitor tacrine (Figure 5). In addition, a series of quinolinone derivatives have been found to show strong AChE inhibition [38,39] such as the natural product harmane [39] and the synthetic compounds T-82 [40] and 4-((2-oxo-1,2-dihydroquinolin-7-yl)oxy)butyl piperidine-1-carbodithioate [41] (Figure 5). Thus, the quinolinone ring system could be a potential key structural point in future optimization for drug development.

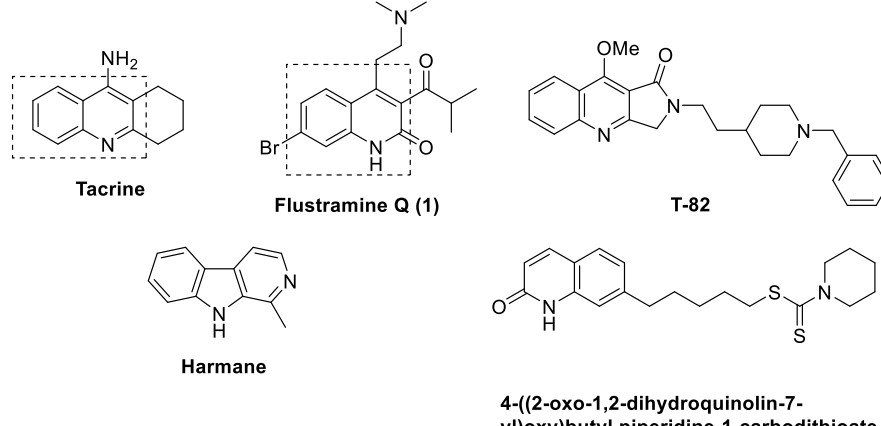

**Figure 5.** Structures of AChE inhibitors structurally related to flustramine Q.

Marine natural products have only been explored to a limited extend as drug leads for treatment of brain dysfunction; however, they include small molecules which can be of interest in neurodegenerative disorders, especially those belonging to alkaloids. It has been shown in this study that flustramine-Q-like marine alkaloids can be potent AChE inhibitors and of interest as future drug leads. Our docking study indicates that simple changes to flustramine Q's structure which restrict the position of the tertiary ammonium group might significantly increase the activity. Further exchange of the bromine with a hydroxy group could improve the activity even more by increasing the number of hydrogen bonds the compound creates with three residues in the binding pocket of AChE. The discovery of flustramine Q, as well as other previously reported AChE inhibitors of marine origin [22,23], suggests that marine natural products' chemical scaffolds should be seriously considered in the search for new therapies in neurodegenerative diseases.

Many approaches for the treatment of AD have been explored in the last couple of decades with limited success [8,42,43]. AChE inhibitors, along with one NMDA receptor antagonist, remain the principal symptomatic treatment for AD and do provide limited therapeutic benefits to patients [7,44]. Although a few AChE inhibitors are present on the market, not all drugs are well tolerated by the patients. New chemical entities of AChE inhibitors with improved activity-adverse effect profiles would give an alternative treatment option for patients and could be combined with other medications.

In conclusion, the marine natural product flustramine Q is a new chemical scaffold for AChE inhibitors. Molecular modelling and docking studies with two in silico designed derivatives of flustramine Q in the binding site of the AChE enzyme indicate increased potency of these derivatives compared with flustramine Q. Flustramine Q possesses the potential to be explored further as a drug lead compound for AD or other neurodegenerative diseases.

**Author Contributions:** Conceptualization, N.M.K. and S.O.; methodology, N.M.K. and X.D.; data curation, N.M.K. and X.D.; formal analysis, N.M.K.; writing—original draft preparation, N.M.K.; writing—review and editing, N.M.K., X.D., S.O. and E.S.O.; visualization, N.M.K.; supervision, E.S.O.; project administration, N.M.K. and E.S.O.; funding acquisition, E.S.O. All authors have read and agreed to the published version of the manuscript.

**Funding:** This research was funded by the Icelandic Research Fund (grant number: 152604051).

**Institutional Review Board Statement:** Not applicable.

**Informed Consent Statement:** Not applicable.

**Data Availability Statement:** All data from in vitro and in silico experiments related to this research work have been generated at the University of Iceland, Faculty of Pharmaceutical Sciences, and have been reported accordingly in the manuscript. The data presented in this study are available on request from the corresponding author.

**Conflicts of Interest:** The authors declare no conflict of interest.

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
