# Peer review of "Flustramine Q, a Novel Marine Origin Acetylcholinesterase Inhibitor from Flustra foliacea"

_futurepharmacol, doi:10.3390/futurepharmacol3010003_

Round 1

Reviewer 1 Report

The work is interesting because it contributes to a current topic with new contributions, from an experimental and also theoretical approach.

The following improvements are proposed:

1.      Since the abstract and the introduction start by talking about AD, it seems to be the main character of this work, however it does not match the title nor the experimental development, nor what is specifically demonstrated. If this were the aim, these parts should be reoriented, for example by proposing them in the title as potential compounds for AD, adding in the IC50 table values of drugs against AD that are AChE inhibitors, taking other complementary targets for docking that account for potential activity for the disease, as there are many targets involved beyond AChE. So I think it is best to remove some of the focus on AD in the initial part of the paper, since at the end of the discussion it is mentioned in the same way to give a final closure to the usefulness of these compounds.

 2.      The IC50 value of the reference, where was it obtained from? It would be convenient to reference it in the table, and even better to be able to measure it to validate the experimental method by comparing it with the reference reported.

 3.      In the methodology it would be proper to cite each of the Schrödinger tools correctly, and to mention protein preparation wizard, which must have been the module used for the preparation of the target.

 4.      The docking score is not well presented as an isolated result, but rather as a point of comparison as used in the discussion, either with other compounds or with the other poses of the same compound.

 5.      In some parts of the discussion that mention improvements in interactions with respect to the new derivatives studied in silico, it is identified what these improvements are, however, in some descriptions only "improvements" are mentioned. It would be helpful to specify whether it corresponds to a particular type of interaction, to the number of interactions, or to the type of interactions.

 6.      In Figure 3 it is recommended to label the amino acid residues to facilitate visual inspection of the interactions, as well as the name of each compound within the figure and also a color legend of the interactions presented.

 7.      In Figure 3.D the amino acid residues are not present, so if the idea is just to compare the poses of each compound, you could use an RMSD, colors that are better distinguished, remove the interactions or add the residues as in the rest of the schemes to clarify the representation.

 8.      In Figure 5, just to facilitate visibility it would be useful to label Flustramine Q as such and also put (1) if necessary, for those who are not familiar with the structure, or only see the images before reading the full text.

 9.      An update of references is needed as there are many old ones. In this regard, you will find updated information regarding approved drugs for AD. Remember that tacrine was withdrawn from FDA (so it may not be a good idea to use it as a structural comparison, but rather the reports of its derivatives that have shown better results) and that there is a new monoclonal antibody available. You can find it in the recent published review https://doi.org/10.3390/ pharmaceutics14091914

10.   Check the mention of figure 4 and figure 5 in the text, it seems to have inconsistencies.(Line 188)

 11.   If in the abstract you only want to talk about AChE inhibitors for AD, perhaps you should specify that point, since there are more than 3 drugs available for AD.

Author Response

On behalf of all the authors I thank the editor of Future Pharmacology and the reviewers for their good comments and constructive criticism of the manuscript. We have addressed all the comments in this rebuttal letter and corrected the manuscript accordingly. The references have been checked and corrected and the English improved. All the changes have been added to the revised manuscript with track changes.

The authors responses to the reviewers’ comments are listed below:

Reviewer 1:

The work is interesting because it contributes to a current topic with new contributions, from an experimental and also theoretical approach.

The following improvements are proposed:

1. Since the abstract and the introduction start by talking about AD, it seems to be the main character of this work, however it does not match the title nor the experimental development, nor what is specifically demonstrated. If this were the aim, these parts should be reoriented, for example by proposing them in the title as potential compounds for AD, adding in the IC50 table values of drugs against AD that are AChE inhibitors, taking other complementary targets for docking that account for potential activity for the disease, as there are many targets involved beyond AChE. So I think it is best to remove some of the focus on AD in the initial part of the paper, since at the end of the discussion it is mentioned in the same way to give a final closure to the usefulness of these compounds.

Authors response: The abstract and the introduction part has been adjusted and the focus on AD has been shifted towards AChE inhibitors.

2. The IC50 value of the reference, where was it obtained from? It would be convenient to reference it in the table, and even better to be able to measure it to validate the experimental method by comparing it with the reference reported.

Authors response: In Materials and methods 2.1.; it is explained that the compounds are screened… “with physostigmine as a positive control” which means that physostigmine was measured in this study along with the test compounds and its IC50 value was determined. The literature value and reference for physostigmine has been added as a footnote to Table 1.

3. In the methodology it would be proper to cite each of the Schrödinger tools correctly, and to mention protein preparation wizard, which must have been the module used for the preparation of the target.

Authors response: The relevant references have been added. In the methods section it is stated “prepared following a standard Protein Preparation Workflow in the Schrödinger’s software” – this means the same as Protein Preparation Wizard.

4. The docking score is not well presented as an isolated result, but rather as a point of comparison as used in the discussion, either with other compounds or with the other poses of the same compound.

Authors response: The docking scores for the three compounds studied are presented in the text of the result section of the manuscript. The docking scores are used to compare the binding affinity estimated by the docking algorithm. Our focus was on evaluating and compare the interaction of flustramine Q and its two in silico designed derivatives with the active site of the AChE enzyme. Only one pose for each compound was presented.

5. In some parts of the discussion that mention improvements in interactions with respect to the new derivatives studied in silico, it is identified what these improvements are, however, in some descriptions only "improvements" are mentioned. It would be helpful to specify whether it corresponds to a particular type of interaction, to the number of interactions, or to the type of interactions.

Authors response: “Improvements” of interaction stated in the discussion part have been connected to type of interaction a few places. Interactions are described in detail in the results part.

6. In Figure 3 it is recommended to label the amino acid residues to facilitate visual inspection of the interactions, as well as the name of each compound within the figure and also a color legend of the interactions presented.

Authors response: The aa residues in fig. 3 have been labelled. The names of the compounds are now given in bold in the figure legend which makes them easier to spot right away. We would like to keep the compound names in the figure legend instead of moving them within the figure as it would make the figure more crowded. At the bottom of the figure legend there is a colour code for all the interactions presented.

7. In Figure 3.D the amino acid residues are not present, so if the idea is just to compare the poses of each compound, you could use an RMSD, colors that are better distinguished, remove the interactions or add the residues as in the rest of the schemes to clarify the representation.

Authors response: The aa residues have been added to Fig. 3D.

8. In Figure 5, just to facilitate visibility it would be useful to label Flustramine Q as such and also put (1) if necessary, for those who are not familiar with the structure, or only see the images before reading the full text.

Authors response: This has been added to Fig. 5

9. An update of references is needed as there are many old ones. In this regard, you will find updated information regarding approved drugs for AD. Remember that tacrine was withdrawn from FDA (so it may not be a good idea to use it as a structural comparison, but rather the reports of its derivatives that have shown better results) and that there is a new monoclonal antibody available. You can find it in the recent published review https://doi.org/10.3390/ pharmaceutics14091914

Authors response: In this study we focus on structural scaffolds of AChE inhibitors that have been used for AD. Other modes of action than AChE inhibition, are out of the scope of this study. We have accepted the good comment from the reviewers and shifted the focus of the introduction part and abstract away from the AD towards the AChE inhibitors. Most of the references cited in the manuscript are published within the last 10 years. Some are older because AChE inhibitors used as AD drugs were approved decades ago and this is reflected in the reference list.

10. Check the mention of figure 4 and figure 5 in the text, it seems to have inconsistencies. (Line 188)

Authors response: This has been corrected.

11. If in the abstract you only want to talk about AChE inhibitors for AD, perhaps you should specify that point, since there are more than 3 drugs available for AD.

Authors response: The abstract has been adjusted. The focus is now kept on the AChE inhibitors that have been used for AD.

Reviewer 2 Report

The manuscript "Flustramine Q, a novel marine origin acetylcholinesterase  inhibitor from Flustra foliacea"

Discussed the acetylcholinesterase inhibition of previously isolated compounds using Ellman's method. What is the source of the compounds?

Two suggested derivatives (Not synthesized) were docked to the crystal structure downloaded from the PDB and no experimental assay.?

Kindly, check the comments in the pdf file.

Furthermore, the docking of all compounds and the co-crystallized ligand to compare the binding energies and the binding with amino acid residues in the active site. The bond types also for comparison.

Author Response

Dear Editor

On behalf of all the authors I thank the editor of Future Pharmacology and the reviewers for their good comments and constructive criticism of the manuscript. We have addressed all the comments in this rebuttal letter and corrected the manuscript accordingly. The references have been checked and corrected and the English improved. All the changes have been added to the revised manuscript with track changes.

The authors responses to the reviewers’ comments are listed below:

Responses to Reviewer 2:

The manuscript "Flustramine Q, a novel marine origin acetylcholinesterase  inhibitor from Flustra foliacea"

Reviewer 2:  the acetylcholinesterase inhibition of previously isolated compounds using Ellman's method. What is the source of the compounds?

Authors response: The source of the compounds used in this study is the marine bryosoan Flustra foliacea. Their isolation and characterization are described in a previous paper by Di et al. (two of the present authors) and is repeatedly cited in this study.

Reviewer 2: Two suggested derivatives (Not synthesized) were docked to the crystal structure downloaded from the PDB and no experimental assay.?

Authors response: Yes, the two in silico designed derivatives of flustramine Q were prepared and used for docking studies. Comparison with flustramine Q indicates that they might be more potent inhibitors. The suggested derivatives were not synthesized or tested in the in vitro assay. This has now been made clearer in the manuscript text.

Reviewer 2: Kindly, check the comments in the pdf file.

Authors response: This has been done and is addressed in the revised manuscript.

Reviewer 2: Furthermore, the docking of all compounds and the co-crystallized ligand to compare the binding energies and the binding with amino acid residues in the active site. The bond types also for comparison.

Authors response: The most active inhibitor of AChE, flustramine Q, was selected for docking studies to identify important interactions of this alkaloid with the AChE active site. Further two in silico designed derivatives of flustramine Q with potentially stronger interactions with the AChE enzyme, were docked and compared with flustramine Q. It was not the aim or a point of interest to dock all the compounds which had already been tested for activity in the in vitro Ellman assay and showed low or no AChE inhibitory activity. We did not study the interaction of the co-crystallized ligand since this is well known from the literature as stated in the results section 3.2 in the manuscript:“Docking results revealed that flustramine Q creates multiple hydrophobic interactions with the binding pocket of AChE including the aromatic residues, Tyr337 and Trp86 known to be important for binding of known inhibitors [8,30].

Reviewer 2: Why did you select this from PDB?

Authors response: The selection of this crystal structure from PDB was based on having a similar size molecule in the binding site, and a human recombinant structure was chosen over an animal version of the AChE enzyme.

Round 2

Reviewer 2 Report

Thank you for replies to my comments. But still, the binding energy of the co-crystalized ligand (huperzine A) is not mentioned.

Author Response

Reviewer 2 comments (2. Round): „Thank you for replies to my comments. But still, the binding energy of the co-crystalized ligand (huperzine A) is not mentioned“.

Authors response: We would like to emphasize that docking scores represent the results of mathematical functions used to predict overall approximate binding affinity of two molecules. Docking scores (not the binding energies of individual interactions) were used for comparison in this study. The aim of the docking study of flustramine Q and its two in silico derivatives I and II was to compare the resulting docking scores and possible interactions with a known PDB crystal structure of the AChE enzyme. Since a crystal structure of AChE co-crystallized with flustramine Q is not available we used a crystal structure with huperzine A, a well known AChE inhibitor. The co-crystallized ligand huperzine A was not a subject of our study and to present the binding energy of individual interactions of this ligand would not serve a purpose in this study. Therefore we would like to keep the manuscript as it is and not discuss or add the binding energies of huperzine A.  The subject of comparison here was the docking scores (not the binding energy of each interaction) of the derivatives I and II compared to flustramine Q and to rationalize the improved docking scores of the derivatives from additional interactions with the active site as described in section 3.2 and 3.3. as well as in the discussion part of the manuscript.